# Alterations in Rumination, Eating, Drinking and Locomotion Behavior in Dairy Cows Affected by Subclinical Ketosis and Subclinical Acidosis

**DOI:** 10.3390/ani14030384

**Published:** 2024-01-25

**Authors:** Ramūnas Antanaitis, Karina Džermeikaitė, Justina Krištolaitytė, Ieva Ribelytė, Agnė Bespalovaitė, Deimantė Bulvičiūtė, Arūnas Rutkauskas

**Affiliations:** Large Animal Clinic, Veterinary Academy, Lithuanian University of Health Sciences, Tilžės Str. 18, LT-47181 Kaunas, Lithuania; karina.dzermeikaite@lsmu.lt (K.D.); justina.kristolaityte@stud.lsmu.lt (J.K.); ieva.ribelyte@lsmu.lt (I.R.); agne.bespalovaite@stud.lsmu.lt (A.B.); deimante.bulviciute@stud.lsmu.lt (D.B.); arunas.rutkauskas@lsmuni.lt (A.R.)

**Keywords:** precision dairy farming, RumiWatch, behavior

## Abstract

**Simple Summary:**

This study investigates the impact of subclinical ketosis (SCK) and subclinical acidosis (SCA) on dairy cow rumination, eating, drinking and locomotion behavior. It focuses on cows in their second or subsequent lactation, producing an average of 12,000 kg/year milk in their previous lactation. The cows were categorized into three groups, SCK, SCA, and healthy cows, based on the milk fat–protein ratio, blood beta-hydroxybutyrate and clinical examinations by a veterinarian. The results show significant differences between the cows affected by SCK and the healthy cows. SCK leads to a reduction in the milk yield (11.78%), rumination time (17.47%), and various eating and chewing behaviors. SCA is associated with a substantial decrease in Eating Time 2 (ET2) by 36.84% and Eating Chews 2 (EC2) by 38.10% compared to the healthy cows. These findings highlight the influence of SCA on feeding behaviors and chewing activity, with implications for nutrient intake and cow health. Additionally, SCK affects locomotion parameters, leading to a 27.36% reduction in the overall activity levels, as well as reductions in the Walking Time (WT), Other Activity Time (OAT), and Activity Change (AC). Early detection and effective management of SCK are crucial for maintaining dairy cow health and productivity. In conclusion, this research underscores the need for advanced strategies to prevent and manage subclinical metabolic disorders in the dairy industry. Effective management and early detection methods are essential to mitigate the impact of SCK and SCA on dairy cow health and benefit the dairy farming sector.

**Abstract:**

This study delves into the effects of subclinical ketosis (SCK) and subclinical acidosis (SCA) on various parameters related to dairy cow rumination, eating, drinking and locomotion behavior. The research hypothesized that these subclinical metabolic disorders could affect behaviors such as rumination, feeding, and locomotion. A total of 320 dairy cows, with a focus on those in their second or subsequent lactation, producing an average of 12,000 kg/year milk in their previous lactation, were examined. These cows were classified into three groups: those with SCK, those with SCA, and healthy cows. The health status of the cows was determined based on the milk fat–protein ratio, blood beta-hydroxybutyrate, and the results of clinical examinations performed by a veterinarian. The data collected during the study included parameters from the RumiWatch sensors. The results revealed significant differences between the cows affected by SCK and the healthy cows, with reductions observed in the rumination time (17.47%) and various eating and chewing behaviors. These changes indicated that SCK had a substantial impact on the cows’ behavior. In the context of SCA, the study found significant reductions in Eating Time 2 (ET2) of 36.84% when compared to the healthy cows. Additionally, Eating Chews 2 (EC2) exhibited a significant reduction in the SCA group, with an average of 312.06 units (±17.93), compared to the healthy group’s average of 504.20 units (±18.87). These findings emphasize that SCA influences feeding behaviors and chewing activity, which can have implications for nutrient intake and overall cow health. The study also highlights the considerable impact of SCK on locomotion parameters, as the cows with SCK exhibited a 27.36% reduction in the walking time levels. These cows also displayed reductions in the Walking Time (WT), Other Activity Time (OAT), and Activity Change (AC). In conclusion, this research underscores the critical need for advanced strategies to prevent and manage subclinical metabolic disorders within the dairy farming industry. The study findings have far-reaching implications for enhancing the well-being and performance of dairy cattle. Effective management practices and detection methods are essential to mitigate the impact of SCK and SCA on dairy cow health and productivity, ultimately benefiting the dairy farming sector.

## 1. Introduction

In dairy farming, automation is being used more and more to cut costs, including labor costs [1]. The economic realities of growing labor costs in relation to capital costs have an impact on this tendency. Dairy farmers can now manage larger herds with less labor thanks to automated technologies, suggesting that the trend of increasing herd numbers is partly due to the adoption of automated systems [2]. Health issues (such as subclinical ketosis and subclinical acidosis) affect a large percentage of dairy cows and have a negative impact on their performance, welfare, and general health [3]. Precision dairy farming (PDF), which involves the monitoring of behavioral, physiological, or production parameters to detect individual animal disease, estrus, or comfort, is experiencing a growing trend in popularity [2]. The present capabilities of PDF technology encompass the ability to monitor several aspects, such as the lying time, rumination time, walking time levels, temperature, and milk yield [4]. To identify cows with health issues, sensor system data can be used, either independently or in combination with proven health-monitoring methods [2]. However, more research is required to create standards for the detection and avoidance of illnesses using information from an automated health monitoring system [5]. We can infer from the findings of our previous study that there is a strong correlation between locomotor behavior and subclinical ketosis. Cows with subclinical ketosis (SCK) were more likely to alternate between ruminating, feeding, and drinking more frequently. There is a correlation between the start of SCK and lower minimum and maximal ambient temperature. Additional research using a greater number of cows is required to validate these findings [6]. In order to develop criteria for treating and differentiating particular health conditions using data from the automatic health monitoring system (AHMS), more study is necessary [5]. For instance, the animals’ walking behavior may have been impacted by metritis, lameness, misplaced abomasum, mastitis, or retained placenta [7]. Locomotion was often lower in SCK-affected cows than in healthy cows [8].

The RumiWatch (RWS) is a novel gadget that combines a noseband sensor and a pedometer, resulting in a versatile system that demonstrates high utility, applicability, sensitivity, and specificity [9]. The RWS noseband sensor has been designed and successfully tested as a scientific monitoring tool for the automated identification of rumination and feeding behaviors in dairy cows that are fed in stables [9]. While the RWS has the capability to forecast feed intake and assess grazing management, it incurs higher costs compared to alternative precision technologies and necessitates daily upkeep.

The developed and verified RumiWatch noseband sensor serves as an appropriate measurement tool for the automated monitoring of the eating and locomotor behavior of dairy cows [9].

Based on the literature, the hypothesis of the current study was that subclinical ketosis and subclinical acidosis had an impact on rumination, eating, and locomotion behavior parameters. To test this hypothesis, we aimed to investigate changes in the rumination, eating, drinking and locomotion behavior of dairy cows affected by subclinical ketosis and subclinical acidosis.

## 2. Materials and Methods

### 2.1. Cows and Study Design 

This study was conducted from 1 June 2023 to 30 September 2023. The approval number for this study was PK012858. The whole experiment was conducted according to the Lithuanian law on animal welfare and protection at one Lithuanian dairy farm (location: 55.819156, 23.773541). The total number of milking cows on this farm was 1150.

### 2.2. Keeping and Feeding of Experimental Cows

All the cows (*n* = 1150) were housed in free-stall barns and fed with a total mixed ration (TMR) balanced according to nutrient requirements [10] with the NorFor^®^ program (Agro Food Park 15, 8200 Aarhus N, Aarhus, Denmark). Water for drinking was freely available. The cows were provided with feed twice a day, at 06:00 and 18:00, following a standard feeding regimen for high-producing, multiparous cows. The feed composition largely comprised of 25% corn silage, 5% alfalfa grass hay, 20% grass silage, 15% sugar beet pulp silage, 30% concentrate mash, and 5% mineral mixture. The formulation of this dietary plan was specifically designed to adequately fulfill the nutritional requirements of a Holstein cow weighing 500 kg and yielding 37 kg of milk per day. The chemical composition of the ration was determined to be 48.8% dry matter (DM), 28.2% neutral detergent fiber (%of DM)*,* 19.8% acid detergent fiber (%of DM)*,* 38.7% non-fiber carbohydrates (%of DM), 15.8% crude protein (%of DM) and 7.1 MJ/kg DM net energy for lactation (NEL). The cows were subjected to a twice-daily milking routine, which occurred at 05:00 and 17:00, employing a parlor-based milking system. The average body weight of the cows was 500 kg, with a standard deviation of 45 kg. The mean energy-corrected milk output, with a fat content of 4.2% and protein content of 3.6%, per individual cow throughout a single lactation period was recorded as 12,000 kg.

### 2.3. Creation of Experimental Groups

For this study, from all the cows (1150), we selected 320 cows (from 5 to 30 days after calving (on average 23 (±7)), second and more lactation, with an average of 12,000 kg/year milk production in past lactation, breed: German Holstein). For these cows, once per day (from 8 to 10 a.m.), the same veterinarian performed a clinical examination. Every time, the same cows were examined. Based on the results of this examination, the milk fat–protein ratio, and blood beta-hydroxybutyrate, we created three groups. The first one was cows with subclinical ketosis (SCK) (*n* = 112); the second one was cows with subclinical acidosis (SCA) (*n* = 102); and the third one was healthy cows (H) (*n* = 106). 

The SCK group was created from cows that had milk F/P levels higher than 1.5 and blood BHB concentrations higher than 1.2 mmol/L, without any other clinical symptoms of disease after calving [11]. During the whole study, these cows were assigned to this group. The number of cows in this group was 112. 

The SCA group (*n* = 102) was created from cows that had rumen motility occurring five to six times every three minutes, milk F/P levels lower than 1.2, mid-to-severe diarrhea, and nondigestive food parts in the feces. For fiber determination, feces was sieved [12].

For the identification of left abomasal displacement, the left flank was percussed. Additionally, there were no clinical signs of other diseases following calving, such as metritis, lameness, mastitis, displaced abomasum, or indigestion. For all the cows, a vaginal examination was undertaken. 

The group of healthy cows (H) was created from healthy cows; during clinical examination, these cows did not show any symptoms of disease after calving. During the whole study, these cows were assigned to this group. The number of cows in this group was 106.

### 2.4. Registration of Parameters

On the SCA and SCA diagnosis days, there were registered parameters from RWS (Table 1). The RumiWatch sensor (RWS) parameters were registered every hour during the whole investigation period, which started on June 1 with 5 days and continued until 30 days postpartum.

#### 2.4.1. Registration of Rumination, Eating, Drinking and Locomotion Behavior Parameters

A noseband halter with an integrated pressure detector and a liquid-filled pressure tube make up the RumiWatch sensor (RWS). The data recorder in this system, which is mounted on the same halter and kept safe in a plastic box, receives a pressure signal from the pressure sensor. Additionally, there is an acceleration sensor to detect triaxial head movements and a robust memory card holder. The pressure data and acceleration measurements are recorded as binary files at a frequency of 10 Hz. Real-time data collecting is made possible via the wireless data transmitter connecting the halter and the RumiWatch Manager software (RumiWatch Manager 2 software (V. 2.2.0.0)). This RWS software’s fundamental algorithms process the accurate categorization of behavioral elements in 10 Hz pressure data in a variety of temporal summaries from which to choose. The algorithms identify clear-cut pressure peak clusters created by jaw movements and then classify them according to their behavioral characteristics. Rumination and eating parameters (such as the rumination time, eating, drinking, gulps, bolus, chews per minute, chews per bolus, ruminate chews, eating chews, other chews) and locomotion parameters (such as walking time, up and down time, minimum, maximum, activity change), were all recorded via the RWS [9].

#### 2.4.2. To Diagnose Hyperketonemia, Whole-Blood BHB Concentrations Were Tested

Samples were collected during farm visits at the same time as feeding each week, that is, two to four hours after a fresh feed delivery, in order to ascertain the highest BHB concentration currently available. During each sampling, the cows were placed in a headlock or a resting stall while a needle-tipped syringe was used to draw a little amount of blood from the coccygeal vein.

#### 2.4.3. With the Use of an In-Line BROLIS Herd

Using a line milk analyzer (Brolis Sensor Technology, Vilnius, Lithuania), the daily milk fat–protein ratios of every cow were recorded. In the 2100–2400 nm spectral range, this device makes use of a special GaSb extensively tunable external cavity laser-based in-line spectrometer. Throughout the milking cycle, the transmission mode was used to continuously check the flow of milk. The primary constituents’ composition level was ascertained via processing the obtained molecular absorption spectra. The analyzer continually measured each cow’s milk composition during each milking cycle. There was no need for extra reagents or maintenance for this “mini-spectroscope”, which was installed on the milking stalls or on a milking robot in the milk flow.

Every five seconds, the analyzer continually measured the composition of each cow’s milk during the milking process. Single values representing full milking were obtained by averaging the dynamics of fat and protein concentrations using weights derived from milk flow.

#### 2.4.4. Duration of Registration of Parameters 

The cows’ clinical examination started on June 1 with 5 days and followed them until 30 days postpartum. Every time, the same cows were examined. During the clinical examination, the whole-blood BHB concentrations were tested. The RWS parameters were registered every hour. The milk fat–protein ratios of every cow were recorded during each milking and a daily average of the milk fat–protein ratio was calculated. The data were analyzed on the SCK and SCA diagnosis days. The average number of days in milk when SCK and SCA were diagnosed was 23 (±7). 

### 2.5. Statistical Analysis

The SPSS software (SPSS Inc., Chicago, IL, USA) was used for all the statistical analyses (IBM Corp., 2017) IBM SPSS Statistics for Windows, version 25.0 (Armonk, NY, USA). The normal distribution of the indicators was verified using the Shapiro–Wilk normality test. The standard error of the mean (M S.E.M.) plus the mean were used to express the data. The mean values of the variables SCK, SCA, and H were compared using Student’s *t*-test. *p* < 0.05 was considered significant. 

## 3. Results

### 3.1. Descriptive Statistics

We found significant differences in the average milk yield per milking session, other activity time, rumination time, Eating Time 1 and Eating Time 2, other chews, rumination chews, Eating Time Chews 1 and Eating Time Chews 2, drinking gulps, bolus, chews per minute, walking time, temp average, temp minimal, temp maximal and activity change between groups. These data are presented in Table 2.

### 3.2. The Impact of Subclinical Ketosis and Subclinical Acidosis on Rumination, Drinking and Eating Behavior Parameters

Rumination time (RT). We found a significant reduction of 17.5% in the RT among the cows affected by SCK (*p* < 0.001) compared to the healthy cows. The mean RT in the SCK group was 15.2 min/h (±0.5), while in the healthy group, it averaged 18.4 min/h (±0.5), and in the subclinical acidosis (SCA) group, it was 19.4 min/h (±0.7).

Eating Time 1 (ET1). A significant (*p* < 0.001) reduction in ET1 was observed of 16.5% in the SCK compared with the healthy group. The mean ET1 in the SCK group was 3.9 min/h (±0.2), whereas in the healthy group, it averaged 4.7 min/h (±0.2), and in the SCA group, it was 6.1 min/h (±0.7).

Eating Time 2 (ET2). We found significant reductions in ET2 in both the SCA (36.8%) and SCK (28.6%) groups compared with the healthy cows (*p* < 0.001). The mean ET2 in the SCA group was 4.6 min/h (±5.2), in the SCK group, it was 5.25 min/h (±7.8), and in the healthy group, it averaged 7.3 min/h (±8.7).

Other chews (OC). The results of our study showed that the cows that were in SCK had 13.8% significantly lower levels of OC compared with the healthy cows (*p* < 0.01). The mean OC concentration in the SCK group was 151.8 n/h (±5.6), while in the healthy group, it averaged 176.0 n/h (±4.8), and in the SCA group, it measured 171.9 n/h (±6.9).

Rumination chews (RC). In our study, the RC was significantly 16.2% lower in the cows affected by SCK compared with the healthy cows (*p* < 0.001). The mean RC in the SCK group was 981.4 n/h (±35.7), while in the healthy group, it averaged 1170.6 n/h (±31.6), and in the SCA group, it measured 1284.1 n/h (±47.6).

Eating Chews 1 (EC1). Based on our findings, a statistically significant reduction of 19.6% in EC1 was observed in the cows with SCK compared with the healthy cows (*p* < 0.001). The mean EC1 in the SCK group was 312.7 n/h (±22.5), while in the healthy group, it averaged 388.9 n/h (±19.1), and in the SCA group, it measured 470.4 n/h (±35.7).

Eating Chews 2 (EC2). We found significantly (*p* < 0.001) lower EC2 in the cows afflicted with both SCA (38%) and SCK (38.1%) in comparison to the healthy cows. The mean EC2 in the SCA group was 312.1 n/h (±17.9), in the SCK group, it averaged 338.7 n/h (±18.3), and in the healthy group, it measured 504.20 n/h (±18.9).

Drinking gulps (DG). In our research, we found that the cows that were in subclinical ketosis had a drop in DG of 36.3% compared with the healthy cows (*p* < 0.001). The mean DG in the SCK group was 131.9 n/h (±10.6), whereas in the healthy group, it averaged 207.0 n/h (±10.8), and in the SCA group, it measured 214.8 n/h (±18.0).

Bolus (B). In our study, we found a significant decrease of 17.0% in B among the cows affected by SCK compared with the healthy cows (*p* < 0.001). The mean B value in the SCK group was 16.9 n/h (±0.6), while in healthy cows, it averaged 20.4 n/h (±0.5), and in cows with SCA, it measured 21.4 n/h (±0.8). 

Chews per minute (CPM). In our investigation of CPM, we identified a significant decrease of 16.7% in CPM in the cows affected by SCK compared with the healthy cows (*p* < 0.001). The mean CPM in the SCK group was 44.8 n/min (±1.2), whereas in the healthy group, it averaged 53.7 n/min (±1.0), and in cows with SCA, it measured 57.7 n/min (±1.4).

### 3.3. The Impact of Subclinical Ketosis and Subclinical Acidosis on Locomotion Behavior Parameters

Walking time (WT). In our study of cow WT, we observed a significant decrease in the WT levels among the cows within both the SCK (27.4%) and SCA (19.1%) groups compared with the healthy cows (*p* < 0.001). The average WT in the SCK group was 53.2 min/h (±1.6), in the SCA group, it measured 59.3 min/h, while in the healthy group, it averaged 73.3 min/h (±1.7).

Activity change (AC). We found a significant decrease of 13.78% in AC in the cows with SCK compared with the healthy cows (*p* < 0.001). The mean AC in the SCK group was 6.4 n/h (±0.2), whereas in the healthy group, it averaged 7.5 n/h (±0.15), and in the SCA group, it measured 7.2 n/h (±0.2).

Other activity time (OAT). Our findings indicated a significant increase of 19.8% in OAT in the cows with SCK compared with the healthy cows (*p* < 0.001). The mean OAT in the SCK group, was 35.2 min/h (±17.2), while in the healthy group, it was 29.2 min/h (±16.3).

## 4. Discussion

With the advent of automated monitoring for eating, drinking and ruminating behavior, a better understanding of the connection between health and husbandry practices has emerged [13]. In dairy cows, automatic rumination and eating behavior monitoring indicates potential for identifying health problems; when both behaviors are combined, post-partum disease detection rates are increased. With the equipment used in our study, individual rumination behavior is easier to monitor than individual feed intake, making it more practical and, thus, more likely to be used on commercial farms [2]. The present study aimed to investigate the effects of subclinical acidosis and ketosis on parameters related to locomotor behavior, rumination, drinking and eating. We examined changes in feeding, rumination, and locomotor behavior in cows with subclinical acidosis and subclinical ketosis to test this theory.

### 4.1. Alterations in Rumination, Eating Behavior in Dairy Cows Affected by Subclinical Ketosis and Subclinical Acidosis

Subclinical ketosis and subclinical acidosis frequently go undetected until they progress to the clinical stage, despite their widespread occurrence in numerous dairy herds [14]. The financial implications associated with these conditions are notably severe when compared to the relatively inexpensive diagnostic methods available for their early detection [15]. SCK is a frequently encountered issue in lactating dairy cows across various herds, and it leads to diminished milk production and changes in milk quality, resulting in noteworthy economic losses for farmers [15].

We found that the cows affected by SCK showed a 17.5% lower rumination time (duration of ruminating chews, with up to 5 s chew breaks). This finding may be indicative of altered digestive processes and potentially reduced nutrient utilization, which could further contribute to the observed decrease in the milk yield. The rumination time has been researched as a method for detecting parturition and illness in commercial dairy cows, either by itself or in conjunction with other factors [2]. If rumination activity is consistently lower in sicker cows than in healthy cows, this has also been the subject of several investigations [16]. The rumination time has been connected in the past to both clinical and subclinical health problems [5]. In the first week following calving, it was observed that cows with SCK rubbed less than healthy cows [17]. Since rumination behavior is likely to be impacted by changes in eating behavior, it could be a good indicator of metabolic problems [18], particularly in the postpartum period [19]. A decrease in DMI may be linked to metabolic illnesses, and rumination behavior may be a useful biomarker for monitoring these disorders [17]. Dairy cattle exhibit wide variations in the amount of time they spend feeding when compared to other experimental conditions. According to White et al. [20], the eating times ranged from 141 to 507 min daily, with a mean of 284 min.

According to our results, we found a reduction in Eating Time 1 (feeding time with head position down within the chosen summary interval) of 16.5% and in Eating Time 2 (feeding time with head position up within chosen interval) of 28.6 in cows with SCK. Although the various criteria employed in different studies to calculate the eating time may account for some of the heterogeneity, the feed management, DMI, the physical and chemical composition of the meal, and the innate diversity of animals all have a significant impact on the eating time [20]. There seems to be a compensatory relationship between the ruminating and eating time. For dairy cows with unlimited access to feed, Dado and Allen [21] found that the correlation coefficient between the eating time and ruminating time was 0.62, indicating that cows who eat less often ruminate for longer. Due to the numerous interdependent factors, estimating how long cows will chew or ruminate might be a helpful management tool for enhancing cow health, although the accuracy can be poor [22].

We found a reduction of 13.8% in other chews (total amount of mastication chews and fear bites during eating), 16.2% in rumination chews (chewing using the mouth during rumination to mechanically break up regurgitated material into smaller pieces), 19.6% in Eating Chews 1 (number of feeding chews with head position down within chosen summary interval), and 38.1% in Eating Chews 2 (number of feeding chews with head position up within chosen summary interval), 16.7% in chews per minute (chewing movements occurring during rumination following regurgitation per minute), 17.0% in bolus (number of regurgitated boli within the chosen summary interval) in cows with SCK. 

In cows with SCA, we found we found reductions in Eating Time 2 of 36.8% and Eating Chews 2 of 38%. Chewing is directly linked to the amount of physically effective fiber consumed, hence tracking chewing action has also been considered a more practical way to identify high-risk cows early on and assess the amount of structural fiber in cows’ diets. On the other hand, research on dairy cows fed only 40% concentrates and with expectedly healthy rumen has shown lower values (<32 chewing min/kg of DMI). This also holds true when using the number of chews per bolus as a health measure rather than the chewing duration. In fact, because the DMI of individual cows need not be measured, measurements of chewing parameters, such as the chews per bolus, are actually considerably simpler to perform under farm conditions. It is assumed that a healthy cow chews a bolus at least 50 times. In fact, when concentrate levels rise, a number of studies have shown a decrease in both the number of boluses that are regurgitated and the number of ruminating chews per bolus [23,24,25,26]. 

However, when additional chewing parameters were examined, the SARA-susceptible cows had slightly fewer feeding chews and, on average, fewer rumination boluses per hour throughout the day. Additionally, it has been proposed that dairy cows that suffer from SARA change the way they sort their diets, possibly by eating more grass to lessen the severity of their rumen fermentation condition. When cows are fed excessive amounts of concentrates at night, they appear to alter their diurnal rumination pattern, increasing rumination during the day and decreasing it during the night [27]. On the day of diagnosis, there were consistent decreases in rumination activity for each health condition, both within the cow and in comparison to cohorts of healthy mates [28]. Individual rumination periods can be programmed to automatically identify health issues that arise after calving, such as metritis and ketosis. Liboreiro et al. [18] stated that while there may be differences in RT and activity between populations of cows that have already experienced periparturient diseases and those that were healthy, more research is required to determine how RT and activity data may be used to predict the occurrence of such periparturient diseases in individuals before they occur. These findings emphasize the considerable impact of SCA on feeding behaviors and raise important considerations for the management of subclinical acidosis in dairy farming. Detecting and addressing such changes early is crucial to ensuring the well-being and productivity of affected cows.

### 4.2. Alterations in Drinking Behavior in Dairy Cows Affected by Subclinical Ketosis

We found a reduction of 36.3% in drinking gulps (the cumulative number of swallows during the drinking process) in cows with SCK. Paudyal et al. [28] predicted a rise in the drinking time postpartum, reaching a maximum minutes per day in early lactation. Though it may be shorter, the drinking time needs to be carefully monitored because greater or more frequent gulps may result in the consumption of more water.

As a result, we would have anticipated a rise in postpartum drinking, with a maximum of eight minutes per day during early lactation. Shorter drinking times, however, may result in greater or more frequent gulps of water consumed; therefore, the effects of drinking times must be carefully evaluated [29]. Furthermore, a variety of factors might impact the amount of water consumed, including variations in the surrounding temperature, increased water loss from increased milk production, feed consumption, consumption of salt and potassium, dry matter in the diet, physiological parameters, and illnesses [29]. Also, measurements of the drinking duration may vary depending on the type of watering system used. While automatic drinking troughs are generally tiny during dry periods and during calving, early- and mid-lactation cows may be able to drink more water due to their large water troughs. Additionally, Pinheiro et al. [30] showed that when cows had access to a higher, larger trough, they drank more water, took more sips, and spent more time consuming. This, however, is inconsistent with the early lactation cows in our trial having shorter drinking periods. Whether cows drinking from large troughs can ingest more water faster than cows using smaller automatic drinking troughs is debatable. Although it does not definitely mean that early lactation cows consume less water, this could account for their shorter drinking periods.

### 4.3. Alterations in Locomotion Behavior in Dairy Cows Affected by Subclinical Ketosis and Subclinical Acidosis

According to our results, cows with SCK had a decrease of 27.4% in walking time (the total amount of time spent walking in a specific recording period expressed as minutes), 13.78% in activity change (activity changes count within the summary time frame), and an increase of 19.8% in other activity time (the time that cows spend engaged in activities other than feeding, rumination, or specific locomotion behaviors). Regarding SCA, we found a reduction only in waking time of 19.1%.

These findings underscore the intricate relationship between SCK and locomotor aspects, highlighting the importance of the early detection and proactive management of this subclinical metabolic disorder in dairy cattle. Our prior research has unveiled significant disparities in walking time, alterations between cows affected by SCK and their healthy counterparts [6]. Walking time-based automated health monitoring systems are effective instruments for identifying metabolic and intestinal issues in dairy cows [5]. It is necessary to consistently conduct a comprehensive health-monitoring program in order to detect health issues in cows during the early postpartum period [5]. Early disease detection, prior to the appearance of distinct clinical symptoms, may improve the treatment response overall and lessen the detrimental long-term effects of disease on the overall cow health and performance [14]. According to Sturm et al. 2020 [31], cows with SKC move less and are slower than healthy cows. Extended periods of inactivity, or paresis, are typical illness behaviors and are clinical indicators of milk fever [32]. Lying is a crucial aspect of cow comfort and a welfare indicator [33]. Lying behavior is relevant in grazing systems for better management of individual dairy cows [34], and it has recently been identified as an early predictor of health issues in housed systems [35]. When utilizing behavior as a welfare or sickness indicator, quantitative research that focuses on characterizing changes in the behavior of healthy cows is crucial to take into account [36]. In contrast to the lying times reported for housed cows, a study assessing the post-calving lying behavior of a group of grazing cows revealed differences in the daily lying times. It was hypothesized that these differences could be caused by outside factors like the time spent walking to and from the milking parlor and the accessibility of feed [37]. In comparison to healthy cows, Rutherford et al.’s study [38] revealed that cows experiencing subclinical ketosis had lower peak activity and shorter durations in activity clusters linked to the first estrus and first insemination postpartum. This could limit the effectiveness of automated surveillance systems in identifying estrus. Animal metabolism is not typically assessed using walking time, although a small number of studies suggest that it may be utilized as a criterion for the early identification of subclinical ketosis. Other cow diseases (mastitis, metritis, etc.) may have an impact on the walking time [39]. Poulopoulou et al. [40] reported that feeding, ruminating, standing, and lying were shown to have strong connections. When comparing ketotic animals to healthy ones, there was a noticeable decrease in the walking time [8]. Prospective future uses include the early identification of metabolic disorders, which can be anticipated by modifications in feeding and rumination behaviors [5], or lameness, which is linked to changes in walking time [41]. According to recent research, certain animal behaviors, such as shifts in rumination and lengths of time spent standing and lying, are linked to preclinical and clinical illnesses, respectively [5].

Some limitations of the current study are as follows. We investigated short-term effects; we focused on dairy cows affected by subclinical ketosis and subclinical acidosis; we did not investigate possibilities for early diagnosis of such diseases, especially concerning changes in the investigated parameters before diagnosis. Future studies should use this information to create a prediction model through machine learning approaches, with an emphasis on a greater number of SCK, SCA, and healthy cows. Also, in future studies, we suggest focusing on the changes in the investigated parameters before diagnosis and also on the long-term implications for the overall health, reproduction, and longevity of dairy cows. 

## 5. Conclusions

In this study, we analyzed alterations in the rumination, drinking, eating and locomotion behavior in dairy cows affected by SCK and SCA. In the case of SCK, we found a reduction in RT (duration of ruminating chews with up to 5 s chew breaks) of 17.5%, Eating Time 1 (feeding time with head position down within the chosen summary interval) of 16.5%, and Eating Time 2 (feeding time with head position up within chosen interval) of 28.6, in other chews (total amount of mastication chews and fear bites during eating) of 13.8%, in rumination chews (chewing using the mouth during rumination to mechanically break up regurgitated material into smaller pieces) of 16.2%, in Eating Chews 1 (number of feeding chews with head position down within the chosen summary interval), 19.6%; and in Eating Chews 2 (number of feeding chews with head position up within the chosen summary interval), 38.1%; in chews per minute (chewing movements occurring during rumination following regurgitation per minute), 16.7%; and in bolus (number of regurgitated boli within the chosen summary interval), 17.0%. Regarding locomotion behavior, we found a reduction of 27.4% in walking time (the total amount of time spent walking in a specific recording period expressed as minutes), 13.78% in activity change (activity changes count within the summary time frame), and an increase of 19.8% in other activity time (the time that cows spend engaged in activities other than feeding, rumination, or specific locomotion behaviors).

During SCA, we found reductions in Eating Time 2 of 36.8%, Eating Chews 2 of 38%, and waking time of 19.1%.

Our findings suggest that, by using innovative technologies such as RWS, we can see that SCK and SCA had a significant influence on various aspects of dairy cow behavioral changes, including eating, rumination, and locomotion behavior. These data can be used for creating algorithms based on registered eating, rumination, and locomotion behavior parameters. 

## Figures and Tables

**Table 1 animals-14-00384-t001:** During this study, there were registered parameters [9].

Parameter	Description	Registration Source
Rumination time (RT) (min/h)	Duration of ruminating chews, with up to 5 s chew breaks	RWS
Eating time 1 (ET1) (min/h)	Feeding time with head position down within chosen summary interval	RWS
Eating time 2 (ET2) (min/h)	Feeding time with head position up within chosen interval	RWS
Drinking time (DT) (min/h)	Drinking duration, including up to 5 s breaks in between drinks	RWS
Other chews (OC) (n/h)	Total amount of mastication chews and fear bites during eating	RWS
Rumination chews (RC) (n/h)	Chewing using the mouth during rumination to mechanically break up regurgitated material into smaller pieces	RWS
Eating chews 1 (EC1) (n/h)	Number of feeding chews with head position down within chosen summary interval	RWS
Eating chews 2 (EC2) (n/h)	Number of feeding chews with head position up within chosen summary interval	RWS
Drinking gulps (DG) (n/h)	The cumulative number of swallows during the drinking process	RWS
Bolus (B) (n/h)	Number of regurgitated boli within the chosen summary interval	RWS
Chews per minute (CPM) (n/min)	Chewing movements occurring during rumination following regurgitation per minute.	RWS
Chews per bolus (CPB) (n/boli)	Chews made when ruminating in between regurgitating and ingesting one bolus	RWS
Walking time (WT) (min/h)	The total amount of time spent walking in a specific recording period expressed as minutes	RWS
Up time (UT) (min/h)	Time spent eating with the head up	RWS
Down time (DT) (min/h)	Time spent eating with the head lowered	RWS
Activity change (AC) (n/h)	Activity changes count within the summary time frame	RWS
Other activity time (OAT) (min/h)	The time that cows spend engaged in activities other than feeding, rumination, or specific locomotion behaviors	RWS

RWS—RumiWatch sensor.

**Table 2 animals-14-00384-t002:** Descriptive statistics for the investigated biomarkers.

Descriptives									
		N	Mean	Std. Deviation	Std. Error	95% Confidence Interval for Mean		Minimum	Maximum
						Lower Bound	Upper Bound		
Rumination time (RT) (min/h)	SCA ^A^	102	19.40	15.06	0.70	18.03	20.78	0	60
	H ^B^	106	18.37	16.19	0.47	17.44	19.31	0	60
	SCK ^C^	112	15.16 ^A,B^	15.31	0.54	14.10	16.23	0	59
	Total	320	17.51	15.77	0.32	16.88	18.14	0	60
Eating Time 1 (ET1) (min/h)	SCA ^A^	102	6.08 ^B,C^	9.51	0.44	5.22	6.95	0	51
	H ^B^	106	4.74	7.65	0.22	4.29	5.18	0	51
	SCK ^C^	112	3.96 ^A,C^	7.34	0.26	3.45	4.47	0	45
	Total	320	4.74	7.97	0.16	4.42	5.06	0	51
Eating Time 2 (ET2) (min/h)	SCA ^A^	102	4.61 ^B^	5.22	0.24	4.14	5.09	0	30
	H ^B^	106	7.30 ^A,C^	8.66	0.25	6.80	7.80	0	48
	SCK ^C^	112	5.25	7.77	0.27	4.71	5.79	0	44
	Total	320	6.11	7.89	0.16	5.79	6.42	0	48
Drinking time (DT) (min/h)	SCA ^A^	102	0.38	0.81	0.03	0.31	0.46	0	6
	H ^B^	106	0.37	0.74	0.02	0.32	0.41	0	6
	SCK ^C^	112	0.39	0.97	0.03	0.33	0.46	0	14
	Total	320	0.38	0.84	0.01	0.35	0.41	0	14
Other chews (OC) (n/h)	SCA ^A^	102	171.87	147.90	6.87	158.36	185.37	0	860
	H ^B^	106	176.00	164.79	4.85	166.49	185.52	0	1484
	SCK ^C^	112	151.79 ^A,B^	148.57	5.25	141.46	162.11	0	1047
	Total	320	167.21	156.73	3.18	160.95	173.46	0	1484
Rumination chews (RC) (n/h)	SCA ^A^	102	1284.13	1025.77	47.6	1190.44	1377.81	0	4077
	H ^B^	106	1170.64	1074.27	31.62	1108.60	1232.69	0	4090
	SCK ^C^	112	981.35 ^A,B^	1015.95	35.96	910.75	1051.94	0	4068
	Total	320	1129.85	1051.67	21.40	1087.89	1171.82	0	4090
Eating Chews 1 (EC1) (n/h)	SCA ^A^	102	470.35 ^B,C^	768.14	35.69	400.20	540.50	0	3848
	H ^B^	106	388.91	650.20	19.14	351.36	426.47	0	3841
	SCK ^C^	112	312.73 ^A,B^	635.6	22.50	268.56	356.90	0	4064
	Total	320	379.35	671.83	13.67	352.54	406.16	0	4064
Eating Chews 2 (EC2) (n/h)	SCA ^A^	102	312.06	385.84	17.93	276.83	347.30	0	2781
	H ^B^	106	504.29 ^A,C^	641.23	18.87	467.25	541.32	0	3634
	SCK ^C^	112	338.70	516.62	18.28	302.80	374.59	0	2902
	Total	320	412.72	566.36	11.52	390.12	435.32	0	3634
Drinking gulps (DG) (n/h)	SCA ^A^	102	214.77	388.36	18.04	179.30	250.24	0	2593
	H ^B^	106	206.97	365.20	10.75	185.87	228.06	0	2528
	SCK ^C^	112	131.88 ^A,B^	298.69	10.57	111.13	152.64	0	2097
	Total	320	183.65	351.22	7.14	169.64	197.67	0	2593
Bolus (B) (n/h)	SCA ^A^	102	21.44	17.19	0.79	19.87	23.01	0	88
	H ^B^	106	20.41	17.89	0.52	19.38	21.44	0	82
	SCK ^C^	112	16.94 ^A,B^	17.23	0.61	15.74	18.14	0	84
	Total	320	19.46	17.63	0.35	18.76	20.16	0	88
Chews per minute (CPM) (n/min)	SCA ^A^	102	57.69	30.93	1.43	54.87	60.52	0	87
	H ^B^	106	53.74	33.79	0.99	51.79	55.69	0	87
	SCK ^C^	112	44.76 ^A,B^	35.29	1.24	42.31	47.22	0	84
	Total	320	51.53	34.12	0.69	50.17	52.89	0	87
Chews per bolus (CPB) (n/boli)	SCA ^A^	102	5.01	11.85	0.55	3.92	6.09	0	161
	H ^B^	106	4.80	9.59	0.28	4.25	5.35	0	60
	SCK ^C^	112	5.16	13.45	0.47	4.23	6.10	0	233
	Total	320	4.96	11.43	0.23	4.50	5.42	0	233
Walking time (WT) (min/h)	SCA ^A^	102	59.28 ^B^	34.83	1.61	56.10	62.46	0	200
	H ^B^	106	73.26 ^A,C^	58.28	1.71	69.90	76.63	0	408
	SCK ^C^	112	53.21 ^B^	46.10	1.63	50.01	56.42	0	248
	Total	320	63.96	51.38	1.04	61.91	66.01	0	408
Up time (UT) (min/h)	SCA ^A^	102	9.72	18.03	0.83	8.07	11.36	0	60
	H ^B^	106	10.24	18.89	0.55	9.15	11.33	0	60
	SCK ^C^	112	8.24	17.76	0.62	7.01	9.48	0	60
	Total	320	9.48	18.37	0.37	8.75	10.21	0	60
Down time (DT) (min/h)	SCA ^A^	102	20.25	22.36	1.03	18.21	22.29	0	60
	H ^B^	106	18.39	21.75	0.64	17.13	19.65	0	60
	SCK ^C^	112	17.44	22.07	0.78	15.90	18.97	0	60
	Total	320	18.43	21.99	0.44	17.55	19.31	0	60
Activity change (AC) (n/h)	SCA ^A^	102	7.15	4.89	0.22	6.71	7.60	0	15
	H ^B^	106	7.47	5.35	0.15	7.16	7.78	0	15
	SCK ^C^	112	6.37 ^A,B^	5.47	0.19	5.99	6.75	0	15
	Total	320	7.05	5.33	0.10	6.83	7.26	0	15
Other activity time (OAT)	SCA ^A^	102	29.17	14.26	0.66	27.86	30.47	0	60
	H ^B^	106	29.21	16.27	0.47	28.27	30.15	0	60
	SCK ^C^	112	35.23 ^A,B^	17.24	0.61	34.03	36.42	1	60
	Total	320	31.19	16.48	0.33	30.53	31.85	0	60

SCA—cows with subclinical acidosis; H—healthy cows; SCK—cows with subclinical ketosis. N—number of cows. The statistically significant differences in the mean values across groups are shown by the letters ^A, B,^ and ^C^.

## Data Availability

The data provided in this study can be found in the publication.

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
