# Peer review of "Alterations in Rumination, Eating, Drinking and Locomotion Behavior in Dairy Cows Affected by Subclinical Ketosis and Subclinical Acidosis"

_animals, 2024, doi:10.3390/ani14030384_

Round 1

Reviewer 1 Report

Comments and Suggestions for Authors

The papers relies on clinical examination that does not measure rumen pH to diagnose rumen acidosis. There is no justification of the accuracy of this and it would best be termed and at risk of acidosis. 

Comments on the Quality of English Language

Overuse of some terms and variation on how movement activity is described needs correction.

Author Response

Dear Reviewer, 

Authors are very thankful for the comments, which help us to improve the manuscript. All changes proposed have been included in the manuscript and highlighted in yellow and track changes.  

Best Regards, 

Prof. Ramunas Antanaitis 

Detail comments and answers:

Rev: Abstract – _need to include diagnostic criteria as without them findings are not interpretable,

Line 47 – _What is the behaviour change a change from and to? Need to explain what it is or delete as not important enough to be in abstract.

Authr: Deleted

Rev: Line 74-75 reword to improve English.

Authr: We corrected to – “Data can often be more helpful for earlier diagnosis and action if they can forecast risk or detect disease more accurately than observations using clinical symptoms”  

Rev: Line 80 - and correct to and

Authr: Corrected to – “and”

Rev: Line 115 – _what is a proper ventilation system – _describe or delete

Authr: Deleted and corrected to – “All cows (n=1150) were housed in free-stall barns, and their dietary intake consisted of a total mixed ration (TMR) that was carefully formulated to meet their physiological requirements consistently throughout the year”

Rev: Line 123 and line 134 – _explain why cows fed for 10% below actual weight? Calculate feed deficit and show in this section.

Authr: This feed was calculated by a farm feed adviser, and we state only facts. Also, the aim of the current study was not to find some impact of feeding on SCK, or SCA; we aimed to investigate changes in rumination, eating, and locomotion behavior in cows with subclinical ketosis and subclinical acidosis.

Rev: Line 137-138 – _duplication of 131 – _delete 131 and only put here.

Authr: Deleted

Rev: Line 150 – _No rumen pH measurement taken. Signs could be infectious disease such as salmonella? “A_v_e_r_a_g_e_ _r_e_c_t_a_l_ _t_e_m_p_e_r_a_t_u_r_e_ _i_s_ _n_o_t_ _s_p_e_c_i_f_i_c_ _e_n_o_u_g_h_._ _I_s_ _c_r_i_t_e_r_i_a_ _t_h_a_t_ _r_e_c_t_a_l_ _t_e_m_p_e_r_a_t_u_r_e_ _w_a_s_ _b_e_l_o_w_ _38.8? Suggest if pH was not measured (it is easy enough to get a rumen sample and to measure pH) then this could be described as nutritional diarrhoea -as that is the main clinical sign.

Authr: We corrected this information following and added reference – “The SCA group (n=102) was created from cows that had rumen motility occurring five to six times every three minutes, milk F/P levels lower than 1.2, mid-to-severe diarrhea, and nondigestive food parts in the feces. For fiber determination, feces were sieved [12].

For identification of left abomasal displacement, the left flank was percussed. Additionally, there were no clinical signs of other diseases following calving, such as metritis, lameness, mastitis, displaced abomasum, or indigestion. For all cows, a vaginal examination was undertaken”

Rev: More detail of clinical examination is needed. Was faeces sieved to determine fibre? Was left flank percussed to determine LDA? As vaginal examination undertaken or just look at discharge on the tail?

Authr: We added information – “For fibre determination, feces were sieved. For identification of left abomasal displacement, the left flank was percussed. For all cows, a vaginal examination was undertaken”

Rev: Line 164 – _is temperature of the environment of the cow?

Authr: We deleted – “Temp average (TA);  Temp minimal (TMin);  Temp maximal (TMax);” because it was a mistake, we didn’t investigate these parameters in this study. 

Rev: Add reference to validation study that RWS is actually measuring water intake etc.

Authr: We added this information – “Table 1. During this study, there were registered parameters [9]”

Rev: Table 1 Activity change – _is this compared to the previous hour or an average of several previous hours?

Authr: Corrected to – “Change in the total amount of time spent walking in a specific recording period expressed as minutes compared to the previous hour”

Rev: Line 171 – _was this a manual or automatic milking system?

Authr: We corrected to – The DeLaval manual milking system (DeLaval Inc., Tumba, Sweden) was used to milk the cows. For milk yield recording DeLaval DelPro™ Farm Management software (DeLaval Inc., Tumba, Sweden) was used

was used.

Rev: Line 188 reference to validation studies for each RWS parameter.

Authr: We added information – “Rumination, eating, drinking, gulps, bolus, chews per minute, chews per bolus, activity up and down time, minimum, maximum, activity change, other chews, ruminate chews, eating chews, and rumination time were all recorded by RWS [9]’

Rev: Line 195 – _reword “A significance level of 0.05 was considered (p < 0.05)”. t_o_ _“P_ _<_ _0_._0_5_ _w_a_s_ _c_o_n_s_i_d_e_r_e_d_ _s_i_g_n_i_f_i_c_a_n_t_”._ _

Authr: Corrected to – “P<0.05 was considered significant”

Rev: There is no information regarding taking blood samples and how they are processed. This needs to be added. Is it at a set time after feeding etc?

Authr: We added information – “2.4.3. To diagnose hyperketonemia, whole-blood BHBA concentrations were tested.  Samples were collected during farm visits at the same time as feeding each week, that is, two to four hours after a fresh feed delivery, in order to ascertain the highest BHBA concentration currently available. During each sampling, the cows were placed in a headlock or a resting stall while a needle-tipped syringe was used to draw a little amount of blood from the coccygeal vein.

Rev: Line 20_5_ _a_d_d_ _“b_e_t_w_e_e_n_ _g_r_o_u_p_s_._ _” _t_o_ _t_h_e_ _e_n_d_ _o_f_ _t_h_e_ _s_e_n_t_e_n_c_e_._ _

Authr: We added and corrected this sentence – “We found significant differences in milk yield, milk lactose, other activity time, rumination time, eating time 1 and eating time 2, other chews, rumination chews, eating time chews 1 and eating time chews 2, drinking gulps, bolus, chews per minute, activity, temp average, temp minimal, temp maximal and activity change between groups”

Rev: Line 213 – _Use data to one decimal place at most as data far less accurate that justifies using 2 decimal places.

Authr: Corrected

Rev: Figure 1 – _data are actual milk yield not change in milk yield,. It represents differences in milk yield between groups not changes from before and after animals became acidotic or ketotic etc. Reword. Also add error bars to all figures and indication of significant differences using superscripts above column and an explanation in the legend.

Authr: Table 2 and Figure 1-15 express the same results, and we deleted figures 1–15 and left only Table 2 and the description of this table.

Rev: Line 221 – _compared to normal? Needs rewording. Reduce repetition and make clear what comparisons are being made. For example, all of line 230 could be removed with no reduction in

clarity. Do not need a title then and explanation of what the next sentence is and then the next sentence. Waste of journal space. Needs editing before sending to review.

Authr: Corrected whole results section.

Rev: S_u_g_g_e_s_t_ _p_u_t_ _d_a_t_a_ _f_o_r_ _s_p_e_c_i_f_i_c_ _a_c_t_i_v_i_t_i_e_s_ _f_i_r_s_t_ _a_n_d_ _“o_t_h_e_r_ _a_c_t_i_v_i_t_y_” _w_h_i_c_h_ _i_s_ _a_ _c_a_t_c_h_ _a_l_l_ _c_a_t_e_g_o_r_y_ _l_a_s_t_._ _

Authr: Corrected – moved to end in section- “3.3. The impact of subclinical ketosis and subclinical acidosis on locomotion behavior parameters”

Rev: L_i_n_e_ _2_4_9_ _r_e_p_l_a_c_e_ _“with a p-value less than 0.001” _w_i_t_h_ _“(_p < 0.001). Do similar throughout. Edit all results in manner indicated by above comments.

Authr: Corrected

Rev: Line 344 – _correct spelling mistakes chuw to chew.

Authr: Corrected.

Rev: Line 356 – _States study looked at locomotor behaviour but none has been presented. Edit to discuss only data actually presented.

Authr: We corrected whole discussion section and added new section – “4.3. Alterations in locomotion behavior in dairy cows affected by subclinical ketosis and subclinical acidosis”

Rev: Line 360 onwards. Results should not be reiterated in the discussion. Delete and just talk about changes and compare to previously published data.

Authr: Deleted  

Rev: Line 405 – _remove results. Just refer to fact they were lower – _not give them again.

Authr: We corrected to – “In the context of subclinical acidosis, our study found significant reductions in Eating Time 2 and Eating Chews 2”  

Rev: Line 418 – _420 needs a reference of the studies quoted.

Authr: We added information – “In fact, when concentrate levels rise, a number of studies have shown a decrease in both the number of boluses that are regurgitated and the number of ruminating chews per bolus [22], [23], [24], [25]”

Rev: Line 442 – _this assumes overall activity – _locomotion. Need to reference validation of that for the s_y_s_t_e_m_ _u_s_e_d_._ _T_a_b_l_e_ _1_ _d_e_s_c_r_i_b_e_s_ _“o_t_h_e_r_ _a_c_t_i_v_i_t_y_ _a_s_ _w_h_e_n_ _n_o_t_ _u_n_d_e_r_g_o_i_n_g_ _l_o_c_o_m_o_t_i_o_n_._ _W_h_e_r_e_ _i_s_ _the locomotion data? Need to decide if walking is going to be called walking, locomotion or Activity and reference the validation of the measure as being actual walking and then stick to one term rather than using them randomly / interchangeably.

Authr: We identified locomotion as: activity up and down time, minimum, maximum, activity change

And corrected to – “Rumination and eating parameters (such as rumination time, eating, drinking, gulps, bolus, chews per minute, chews per bolus, ruminate chews, eating chews, other chews) and locomotion parameters (such as activity up and down time, minimum, maximum, activity change,) were all recorded by RWS [9]”

Rev: Line 450 – _reference prior research

Authr: We corrected to – “Our prior research has unveiled significant disparities in activity alterations between cows affected by SCK and their healthy counterparts [6]”

Rev: Line 458 – _H_o_w_ _w_a_s_ _“ _a_t_ _r_i_s_k_ _f_o_r_ _S_K_C_” _d_e_f_i_n_e_d_ _i_n_ _t_h_i_s_ _p_a_p_e_r_ _v_e_r_s_e_s_ _y_o_u_r_ _d_i_a_g_n_o_s_i_s_ _o_f_ _S_K_C_?_ _

Authr: We corrected to – “According to Sturm et al. 2020 [24], cows with SKC move less and are slower than healthy cows”

Rev: Line 470- 482 – _speculative - suggests delete.

Authr:  Deleted

Rev: Line 499 – _502 – _Remove result reiteration.

Authr: We deleted this paragraph

Rev: Line 49 – _514 – _Just results again so delete all. No discussion.

Authr: We deleted this paragraph.

Rev: Were is the detail of the lactose data?

Authr: We deleted information about lactosis concentration because we did not investigate it in this study. 

Rev: How are eating time 1 and eating time 2 different? Table 1 gives them the same definition. Also check definition of Eating Chew 1 and 2.

Authr: We corrected  - Table 1. During this study, there were registered parameters [9] and added following definitions-

Eating time 1 (ET1) (min/h)

Feeding time with head position down within chosen summary interval

Eating time 2 (ET2)

(min/h)

Feeding time with head position up within chosen interval

Eating chews 1 (EC1)

(n/h)

Number of feeding chews with head position down within chosen summary interval

Eating chews 2 (EC2)

(n/h)

Number of feeding chews with head position up within chosen summary interval

Bolus (B) (n/h)

Number of regurgitated boli within the chosen summary interval

Rev: I_s_ _“b_o_l_u_s_” _o_n_l_y_ _d_r_i_n_k_i_n_g_ _o_r_ _d_o_e_s_ _i_t_ _i_n_c_l_u_d_e_ _e_a_t_i_n_g_ _o_r_ _r_e_g_u_r_g_itation?

Authr: We added information – “Bolus (B) (n/h) - Number of regurgitated boli within the chosen summary interval

Rev: Conclusion – _t_o_o_ _m_u_c_h_ _u_s_e_ _o_f_ _t_e_r_m_ _“u_n_d_e_r_s_c_o_r_e_”._ _R_e_w_o_r_d_ _t_o_ _u_s_e_ _s_o_m_e_t_h_i_n_g_ _s_c_i_e_n_t_i_f_i_c_ _a_n_d_ _meaningful.

Authr: We corrected conclusion section following – “In this study, we analyzed alterations in rumination, drinking eating and locomotion behavior in dairy cows affected by SCK and SCA. In the case of SCK, we found a reduction in RT (duration of ruminating chews with up to 5-second chew breaks) of 17.5%, eating time 1 (feeding time with head position down within the chosen summary interval) of 16.5%, and eating time  2 (feeding time with head position up within chosen interval) of 28.6, in other chews (total amount of mastication chews and fear bites during eating) of 13.8%, in rumination chews (chewing using the mouth during rumination to mechanically break up regurgitated material into smaller pieces) of 16.2%, in eating chews 1 (number of feeding chews with head position down within the chosen summary interval), 19.6%; and in eating chews 2 (number of feeding chews with head position up within the chosen summary interval), 38.1%; in chews per minute (chewing movements occurring during rumination following regurgitation per minute), 16.7%; and in bolus (number of regurgitated boli within the chosen summary interval), 17.0%. Regarding locomotion behavior, we found a reduction of 27.4% in walking time (the total amount of time spent walking in a specific recording period expressed as minutes), 13.78% in activity change (activity changes count within the summary time frame), and an increase of 19.8% in other activity time (the time that cows spend engaged in activities other than feeding, rumination, or specific locomotion behaviors).

During SCA, we found reductions in eating time 2 of 36.8%, eating chews 2 of 38%, and waking time of 19.1%.

Our findings suggest that, by using innovative technologies such as RWS, we can see that SCK and SCA had a significant influence on various aspects of dairy cow behavioral changes, including eating, rumination, and locomotion behavior. These data can be used for creating algorithms based on registered eating, rumination, and locomotion behavior parameters”

Rev: Need discussion of limitations of the study added.

Authr: We added in discusion section – “Some limitations of the current study are: we investigated short-term effects; we focused on dairy cows affected bysubclinical ketosis and subclinical acidosis; but we didn’t investigate possibilities for early diagnosis of such diseases, especially about changes in investigated parameters before diagnosis. Future studies should use this information to create a prediction model through machine learning approaches, with an emphasis on a greater number of SCK, SCA, and healthy cows. Also, in future studies, we suggest focusing on the changes in investigated parameters before diagnosis and also on the long-term implications for the overall health, reproduction, and longevity of dairy cows”

Reviewer 2 Report

Comments and Suggestions for Authors

The purpose of this manuscript is investigate changes in in-line milk lactosis concentration and rumination, eating, and locomotion behavior in cows with subclinical ketosis and subclinical acidosis. Please find below some suggestions to improve the paper.

1. Lack of comparison with traditional methods, unable to show the advantages of this early detection.

2. This manuscript mainly focuses on physiological characteristics, and establishes the relationship between physiological characteristics and subclinical acidosis and subclinical ketosis. However, the results do not show the correlation analysis between physiological characteristics and metabolic diseases.

3. Table 2 and Figure 1-15 express the same results, why do you want to repeat them?

4. The results shown in Figures 16-18 are all related to physiological features, and their correlation cannot verify the main idea of this manuscript.

Author Response

Dear Reviewer,
We sincerely appreciate your thorough review of our manuscript, and we are grateful for the valuable comments that have significantly contributed to the enhancement of the quality of our work. We have carefully addressed each of your suggestions and made the necessary corrections. Below, we provide detailed responses to the points raised in your review:

Reviewer .: Lack of comparison with traditional methods, unable to show the advantages of this early detection.

Authors: We focus not on early diagnosis, but on changes in rumination, eating, drinking and locomotion behavior in dairy cows affected by subclinical ketosis and subclinical acidosis
We corrected materials and methods section– “ For this study, from all cows (1150), we selected 320 cows (from 5 to 30 days after calving (on average 23 (±7)), second and more lactation, with an average of 12,000 kg/year milk production in past lactation, breed: German Holstein). For these cows, one per day (from 8 to 10 a.m.), the same veterinarian performed a clinical examination. Every time, the same cows were examined. Based on the results of this examination, milk fat-protein ratio, and blood beta-hydroxybutyrate, we created three groups. The first one is cows with subclinical ketosis (SCK) (n = 112); the second one is cows with subclinical acidosis (SCA) (n = 102); and the third one is healthy cows (H) (n = 106). The SCK group was created from cows that had milk F/P levels higher than 1.5 and blood BHB concentrations higher than 1.2 mmol/L without any other clinical symptoms of disease after calving [11].  During the whole study, these cows were assigned to this group. The number of cows in this group was 112. 
The SCA group (n=102) was created from cows that had rumen motility occurring five to six times every three minutes, milk F/P levels lower than 1.2, mid-to-severe diarrhea, and nondigestive food parts in the feces. For fiber determination, feces were sieved [12].
For identification of left abomasal displacement, the left flank was percussed. Additionally, there were no clinical signs of other diseases following calving, such as metritis, lameness, mastitis, displaced abomasum, or indigestion. For all cows, a vaginal examination was undertaken. 

Also, we corrected aim of this study – “Based on the literature, the hypothesis of the current study was that subclinical ketosis and subclinical acidosis had an impact on, rumination eating, and locomotion behavior parameters. To test this hypothesis, we aimed to investigate changes in rumination, eating, drinking and locomotion behavior in dairy cows affected by subclinical ketosis and subclinical acidosis” 

Reviewer: This manuscript mainly focuses on physiological characteristics, and establishes the relationship between physiological characteristics and subclinical acidosis and subclinical ketosis. However, the results do not show the correlation analysis between physiological characteristics and metabolic diseases.

Authors: In this study, we did not focus on correlation analysis between physiological characteristics and metabolic diseases. In this study, we analyzed alterations in rumination, drinking eating and locomotion behavior in dairy cows affected by SCK and SCA. We made more clary methodology, discussion and conclusion section 

Reviewer: Table 2 and Figure 1-15 express the same results, why do you want to repeat them?

Authors: Thanks for suggestion, we deleted these figures, and corrected results section. 

Reviewer: The results shown in Figures 16-18 are all related to physiological features, and their correlation cannot verify the main idea of this manuscript.

Authors: We deleted these figures, because they don’t support our aim.

Reviewer 3 Report

Comments and Suggestions for Authors

The manuscript ANIMALS 2725453, entitled “Alterations in Rumination and Eating Behavior in Dairy Cows Affected by Subclinical Ketosis and Subclinical Acidosis” deals with an interesting issue but the design of the study (Materials and Methods) remains quite unclear and the manuscript itself is poorly prepared. Therefore, it cannot be considered for publication in its present form. Several specific comments (the most important ones) follow:

SIMPLE SUMMARY

L12: No data on milk composition are presented.

L14: Subclinical diseases diagnosed based on clinical examinations by a veterinarian?

ABSTRACT:

L28: Again, milk composition! No data included in the text submitted.

L29: “in-line milk lactose concentration”; why only lactose? This is the least interesting of milk ingredients!

L33-35: Again, clinical examinations and specific diagnostic criteria, for subclinical diseases!

L36: What do you mean with “DeLaval milking system”? Was it an automatic milking system (a “robot”)? Later in the text, we find out it wasn’t. Not needed here; delete.

KEY WORDS:

L55: Both “innovation in dairy cattle” and “diseases after calving” are too vague; “early diagnosis” is not applicable since such data or discussion is not provided.

INTRODUCTION:

L62-64: Subclinical acidosis is not an early post-partum disease, only; syntax suggests so. Please, rephrase and move affect after the parentheses. Try not to use “affect” twice in the same sentence.

L68: Is “laying time” a correct term? I believe it is “lying time”.

L69-71: Obviously, this is an example, but does it fit with the context of this paper? Did you proceed to early diagnosis of SCK and SCA?

L73-76: A continuation of the previous comment; if you are not providing such data here, you must “connect” your work with these issues. And “better” is certainly preferable to “more well”.

L80-81: Subclinical acidosis was not included in you previous publication.

L83: “quantity of cows” or “number of cows”?

L86-89: Are there such data in reference [9]?

L95-97: If this is the case (“rather than practical use within the dairy farming industry), how useful is your research? Perhaps, you want to rephrase this.

L97-100: No need to repeat this one more time; you have already done so.

L102-106: Does this correspond with the rest of the study (Materials and Methods, Results)?

MATERIALS AND METHODS:

L111: “practical” is probably not a good choice, here.

L116: “their dietary intake consisted of a total mixed ration (TMR)” is not a good phrase, either.

L117: “to meet their physiological requirements” could trigger a long discussion (the “physiological” part of it). “Nutrient requirements” is more neutral. You have to declare, though, under which nutritional system those requirements were met.

L121: “grain” is farmers’ vocabulary and not accurate for a scientific paper. Oil-meals were most probably included (and not only grains).

L123: Later on, we learn that these were German Holsteins (not “Germany” L140); we also learn that animals monitored had a BW of 550 kg. Why were rations formulated for 500 kg cows? Moreover, I would certainly love to see cows of that size and weight producing 12,000 kg of milk (L135-136). Is there a mistake here?

L124-125: Dry matter does not account for whatever part of composition. Keep your English simple.

L128-129: “and net lactation energy was measured at Mcal/kg. The numerical value 1.6”. Please, rephrase.

L129: “Bovine animals”? Not just cows?

L132-133: You most certainly mean “parity”, here.

L133-134: “time frame of 5 to 30 days following calving”. You mean when they were employed in the study. Towards 25-30 days after calving, do you consider this “early post-partum”? Is this in accordance with the references you use at the Introduction section?

L140-142: You must be very careful when using the word “selected”; you had a parity criterion, that’s OK. What do you mean by “selected”? Were there any other cows with SCA or SCK that were not “selected”? In L109 you state that “This study was conducted from June 1, 2023, to September 30, 2023”. Combining this with L140-142, facts become difficult to follow.  Did the veterinarian examine eligible cows ONCE and assigned them in various groups or did he examine all cows every day? For how long did this happen and for how long were the cows “followed” by their data (milk production and RWS, before and after assignment?). Was it the day of assignment only? If the latter is true, how useful are your results? How can we discuss, early disease detection? Overall, I cannot follow your procedures: Did you on June 1, examined all cows in the 5 to 30 DIM windows and assigned them in groups and then all cows reaching DIM 5 were also included? Did you on June 1 started with DIM 5 cows and followed them until DIM 30? Please specify.

L141-144: The way you present it here (and elsewhere) it seems as subclinical disease groups were formed based on clinical examination. You certainly don’t mean this; consider rephrasing.

L145-153: Reference [13] is OK for SCK; is it OK for SCA, as well? You need references for SCA; are clinical observations specific enough? You combined them with F/P<1.2? Please, explain clearly and in detail. Cows assigned to a group remained to that group for the whole study period. Again, how long was the study period? What if F/P levels changed during the study period? Data collected apply to which period? How was Fat and Protein content measured? When? What were the Somatic Cell Scores among groups? Do you have SCS? How did you get them?

L158-164: These are included in Table 1; no need to report them in the text.

L168: TABLE 1. a) “Milk yield per milking session (kg)”: I can understand it, as separate recordings. For how many days were data available? b) What does “specific locomotion behaviors” mean? c) Eating times 1 and 2 have the same descriptions. d) What are “fear bites”? e) Eating chews 1 and 2, check definitions. f) Bolus definition? Isn’t bolus used for feed? g) Chews per minute/per bolus definitions are hard to follow. h) Activity and activity change: how do you define “specific recording periods”?

L170-172: You need to name the software used for milk recording, here.

L186: Environmental or cow temperatures?

L194-195: This implies that some variables were not normally distributed; is this true? If yes, how were those variables analyzed? Are SCK, SCA and H really variables? If this is a syntax issue, please be careful!

L197-198: Besides the statistical controversies, do you have such results available? Overall, one has to assume what the statistical analysis was; this is not a clear description.

RESULTS

L202: Lactose?

L207: TABLE 2: The use of superscripts A, B and C is quite unusual, to say the least.

L214-216: Milk yields are obviously per “milking session”. What an unusual way to report milk yields! Still, about how many recordings do we talk about? Please, consider that no lactose data are presented.

L222-225: “In the Subclinical Acidosis (SCA) group, OAT was 20.6% higher than in the healthy group. The mean OAT in the SCK group was 35.23 minutes (±17.24), while in the healthy group, it averaged 29.21 minutes (±16.27), and in the SCA group, it was 29.17 minutes (±14.26)”. Is 29.17 min 20.6% higher than 29.21 min?

L233-235: “The mean RT in the SCK group was 15.16 hours 233 (±0.54), while in the healthy group, it averaged 18.37 hours (±0.47), and in the Subclinical 234 Acidosis (SCA) group, it was 19.40 hours (±0.7)”. Aren’t these rumination times extremely high? Can this be correct? If you add RT, OAT, ET1 and ET2, it is more than 24 hours. Per day, I assume.

L239-335: Without proper definitions for most of them, it is impossible to review them. What is an “activity unit”?

L336-352: No surprise, really, regarding correlations reported. I need the bolus definition.

DISCUSSION:

L356-357: Lactose, again.

L360-366: These are results; delete.

L368-370: Any reference for SCK frequently turning clinical?

L375-403: All these could be OK, if RT recorded were normal. I would not dare to consider a cow ruminating for 15 hours per day as a sick one, even if others were ruminating for 18 hours. For how long were cows found to be ruminating in other studies and with other equipment? What about lactating cow time budgets?

L404-437: This is a text quite difficult to follow. It must be structured correctly.  The sentence in L405-406 makes no sense. SCA and SCK are all mixed up. Without proper definitions on hand, this is an impossible task. Again, what is the “unit” reported? What other researchers report on these issues? How do your results compare to those of others?

L438-451: A long text, just to support findings reported in this and in the previous study. Avoid repeating results. This is the place to report and discuss what others found (not in L484). Shorten and restructure this section.

L451-469: Most of this is “introduction” stuff. Delete or remove. Text in L457-458 does relate to comment above. Early detection of diseases seems to be a recurring issue for the authors.  They provide no such findings themselves, though. What is your point here? Early detection is feasible through behavioral changes? What is your contribution? Do you report changes before any other method of diagnosis?

L470-483: This is an attempt to explain reduced activity. Fair, but with lot of speculation present. Useful only if using literature correctly.

L484-498: Blend this part of the manuscript with the discussion text of corresponding findings.

L499-514: Nothing new, nothing useful. Delete or replace with more relevant discussion.

CONCLUSIONS

L518-521: “In conclusion, our findings strongly support the notion that SCK exerts a significant influence on various aspects of dairy cow health and productivity, including milk production, eating behavior, rumination, body condition, and locomotion”. Did you really find a link between SCK and dairy cow health? What you describe is behavioral changes, not health issues.

L529-531: “Furthermore, this study underscores the necessity for enhanced strategies aimed at preventing and mitigating subclinical metabolic disorders within the dairy farming industry”.  Just because behavior changes?

Comments on the Quality of English Language

Moderate editing is required.

Author Response

Dear Reviewer, 

Authors are very thankful for the comments, which help us to improve the manuscript. All changes proposed have been included in the manuscript and highlighted in yellow and track changes.  

Best Regards, 

Prof. Ramunas Antanaitis 

Detail comments and answers:

SIMPLE SUMMARY

Rev: L12: No data on milk composition are presented.

Auth: We corrected to – “This study investigates the impact of subclinical ketosis (SCK) and subclinical acidosis (SCA) on dairy cow rumination and eating behavior”

Rev: L14: Subclinical diseases diagnosed based on clinical examinations by a veterinarian?

Auth: We corrected this sentence to – “The cows were categorized into three groups: SCK, SCA, and healthy cows, based on milk fat – protein ratio, blood beta-hydroxybutyrate and clinical examinations by a veterinarian”

ABSTRACT:

Rev: L28: Again, milk composition! No data included in the text submitted.

Auth: We corrected this sentence to – “This study delves into the effects of subclinical ketosis (SCK) and subclinical acidosis (SCA) on various parameters related to dairy cow rumination and eating behavior”

Rev: L29: “in-line milk lactose concentration”; why only lactose? This is the least interesting of milk ingredients! 

Auth:We corrected to – “The research hypothesized that these subclinical metabolic disorders could affect behaviors such as rumination, feeding, and locomotion”

Rev: L33-35: Again, clinical examinations and specific diagnostic criteria, for subclinical diseases!

Auth:We corrected to – “The health status of the cows was determined based on milk fat-protein ratio, blood beta-hydroxybutyrate, and the results of clinical examinations performed by a veterinarian”

Rev: L36: What do you mean with “DeLaval milking system”? Was it an automatic milking system (a “robot”)? Later in the text, we find out it wasn’t. Not needed here; delete.

Auth:Deleted and corrected this sentence to – “The data collected during the study included parameters from the Rumiwatch sensors”

KEY WORDS:

Rev: L55: Both “innovation in dairy cattle” and “diseases after calving” are too vague; “early diagnosis” is not applicable since such data or discussion is not provided.

Auth:Corrected to – “precision dairy farming; RumiWatch; behavior”

INTRODUCTION:

Rev: L62-64: Subclinical acidosis is not an early post-partum disease, only; syntax suggests so. Please, rephrase and move affect after the parentheses. Try not to use “affect” twice in the same sentence.

Auth:We corrected this sentence to – “Health issues (such as subclinical ketosis and subclinical acidosis) affect a large percentage of dairy cows and have a negative impact on their performance, welfare, and general health”

Rev: L68: Is “laying time” a correct term? I believe it is “lying time”.

Auth:We corrected to – „lying time“

Rev: L69-71: Obviously, this is an example, but does it fit with the context of this paper? Did you proceed to early diagnosis of SCK and SCA?

Auth:We deleted this sentence

Rev: L73-76: A continuation of the previous comment; if you are not providing such data here, you must “connect” your work with these issues. And “better” is certainly preferable to “more well”.

Auth:We deleted this sentence

Rev: L80-81: Subclinical acidosis was not included in you previous publication.

Auth:We corrected this sentence to – „Cows with subclinical ketosis (SCK) were more likely to alternate between ruminating, feeding, and drinking more frequently. There is a correlation between the start of SCK and lower minimum and maximal ambient temperature. Additional research using a greater quantity of cows is required to validate these findings“

Rev: L83: “quantity of cows” or “number of cows”?

Auth:We corrected to – „number of cows“

Rev: L86-89: Are there such data in reference [9]?

Auth:We added new reference – “For instance, the animals' walking behavior may have been impacted by metritis, lameness, misplaced abomasum, mastitis, or retained placenta [7]” A field investigation of the use of the pedometer for the early detection of lameness in cattle - PMC (nih.gov)

Rev: L95-97: If this is the case (“rather than practical use within the dairy farming industry), how useful is your research? Perhaps, you want to rephrase this.

Auth:We corrected to – „The developed and verified RumiWatch noseband sensor, which serves as an appropriate measurement tool for automated monitoring of the eating and locomotor behavior of dairy cows [11]”

Rev: L97-100: No need to repeat this one more time; you have already done so.

Auth:We deleted these sentences.

Rev: L102-106: Does this correspond with the rest of the study (Materials and Methods, Results)?

Auth:We corrected to – „Based on the literature, the hypothesis of the current study was that subclinical ketosis and subclinical acidosis had an impact on, rumination eating, and locomotion behavior parameters. To test this hypothesis, we aimed to investigate changes in rumination, eating, and locomotion behavior in cows with subclinical ketosis and subclinical acidosis”

MATERIALS AND METHODS:

Rev: L111: “practical” is probably not a good choice, here.

Auth:We corrected to – “The whole experiment was conducted according to the Lithuanian law on animal welfare and protection at one Lithuanian dairy farm (location: 55.819156, 23.773541)”

Rev: L116: “their dietary intake consisted of a total mixed ration (TMR)” is not a good phrase, either.

Auth:We corrected to – “All cows (n = 1150) were housed in free-stall barns and fed with a total mixed ration (TMR) that was carefully formulated to meet their physiological requirements consistently throughout the year”

Rev: L117: “to meet their physiological requirements” could trigger a long discussion (the “physiological” part of it). “Nutrient requirements” is more neutral. You have to declare, though, under which nutritional system those requirements were met.

Auth: Corrected to – “All cows (n = 1150) were housed in free-stall barns and fed with a total mixed ration (TMR) balanced according to nutrient requirements [10] with to the NorFor® program (Agro Food Park 15, 8200 Aarhus N, Aarhus, Denmark) Water for drinking was freely available”

Rev: L121: “grain” is farmers’ vocabulary and not accurate for a scientific paper. Oil-meals were most probably included (and not only grains).

Auth: We corrected to – “..30% concentrate mash..”

Rev: L123: Later on, we learn that these were German Holsteins (not “Germany” L140); we also learn that animals monitored had a BW of 550 kg. Why were rations formulated for 500 kg cows? Moreover, I would certainly love to see cows of that size and weight producing 12,000 kg of milk (L135-136). Is there a mistake here?

Auth: We corrected to – “The average body weight of the cows was 500 kg with a standard deviation of 45 kg”

Rev: L124-125: Dry matter does not account for whatever part of composition. Keep your English simple.

Auth: We corrected this sentence to – “The chemical composition of the ration was determined to be: 48.8% dry matter (DM), 28.2% neutral detergent fiber (%of DM), 19.8% acid detergent fiber (%of DM), 38.7% nonfiber carbohydrates (%of DM), 15.8% crude protein (%of DM) and net lactation energy was measured at Mcal/kg”

Rev: L128-129: “and net lactation energy was measured at Mcal/kg. The numerical value 1.6”. Please, rephrase.

Auth: We corrected to – “…and 7.1 MJ/kg DM net energy for lactation (NEL)”

Rev: L129: “Bovine animals”? Not just cows?

Auth: We corrected to – “The cows were subjected..”

Rev: L132-133: You most certainly mean “parity”, here.

Auth: We deleted this sentence because it was a repetition of the one below.

Rev: L140-142: You must be very careful when using the word “selected”; you had a parity criterion, that’s OK. What do you mean by “selected”? Were there any other cows with SCA or SCK that were not “selected”? In L109 you state that “This study was conducted from June 1, 2023, to September 30, 2023”. Combining this with L140-142, facts become difficult to follow.  Did the veterinarian examine eligible cows ONCE and assigned them in various groups or did he examine all cows every day? For how long did this happen and for how long were the cows “followed” by their data (milk production and RWS, before and after assignment?). Was it the day of assignment only? If the latter is true, how useful are your results? How can we discuss, early disease detection? Overall, I cannot follow your procedures: Did you on June 1, examined all cows in the 5 to 30 DIM windows and assigned them in groups and then all cows reaching DIM 5 were also included? Did you on June 1 started with DIM 5 cows and followed them until DIM 30? Please specify. 

Auth: We corrected information –

“For this study, from all cows (1150), we selected 320 cows (from 5 to 30 days after calving (on average 23 (±7)), second and more lactation, with an average of 12,000 kg/year milk production in past lactation, breed: German Holstein). For these cows, one per day (from 8 to 10 a.m.), the same veterinarian performed a clinical examination. Every time, the same cows were examined. Based on the results of this examination, milk fat-protein ratio, and blood beta-hydroxybutyrate, we created three groups”

“The SCA group (n=102) was created from cows that had rumen motility occurring five to six times every three minutes, milk F/P levels lower than 1.2, mid-to-severe diarrhea, and nondigestive food parts in the feces. For fiber determination, feces were sieved [13].

For identification of left abomasal displacement, the left flank was percussed. Additionally, there were no clinical signs of other diseases following calving, such as metritis, lameness, mastitis, displaced abomasum, or indigestion. For all cows, a vaginal examination was undertaken”

We added new one section – “2.4.4. Duration of registration of parameters The cows clinical examination started on June 1 with 5 days and followed them until 30 days postpartum. Every time, the same cows were examined. During the clinical examination, whole-blood BHB concentrations were tested. The RWS parameters were registered every hour. Milk fat-protein ratios of every cow were recorded during each milking and a daily average of the milk fat-protein ratio was calculated. The data were analyzed on SCK and SCA diagnosis days. The average number of days in milk when SCK and SCA were diagnosed was 23 (±7)”

Rev: L141-144: The way you present it here (and elsewhere) it seems as subclinical disease groups were formed based on clinical examination. You certainly don’t mean this; consider rephrasing.

Auth: We corrected to – “Based on the results of this examination, milk fat-protein ratio, and blood beta-hydroxybutyrate, we created three groups”

Rev: L145-153: Reference [13] is OK for SCK; is it OK for SCA, as well? You need references for SCA; are clinical observations specific enough? You combined them with F/P<1.2? Please, explain clearly and in detail. Cows assigned to a group remained to that group for the whole study period.

Auth: We corrected to – “The SCA group (n=102) was created from cows that had rumen motility occurring five to six times every three minutes, milk F/P levels lower than 1.2, mid-to-severe diarrhea, and nondigestive food parts in the feces. For fiber determination, feces were sieved [13]”

Rev: Again, how long was the study period? What if F/P levels changed during the study period? Data collected apply to which period? How was Fat and Protein content measured? When? What were the Somatic Cell Scores among groups? Do you have SCS? How did you get them?

Auth: We added new section – “2.4.4. Duration of registration of parameters The cows clinical examination started on June 1 with 5 days and followed them until 30 days postpartum. Every time, the same cows were examined. During the clinical examination, whole-blood BHB concentrations were tested. The RWS parameters were registered every hour. Milk fat-protein ratios of every cow were recorded during each milking and a daily average of the milk fat-protein ratio was calculated. The data were analyzed on SCK and SCA diagnosis days. The average number of days in milk when SCK and SCA were diagnosed was 23 (±7)” We didn’t measure SCS during this study.  

Rev: L158-164: These are included in Table 1; no need to report them in the text.

Auth: We deleted this information and corrected to – “During SCA and SCA diagnosis days, there were registered parameters from RWS and DeLaval milking system (Tab. 1). The Rumiwatch sensors (RWS) parameters were registered every hour during the whole investigation period.”

Rev: L168: TABLE 1. a) “Milk yield per milking session (kg)”: I can understand it, as separate recordings. For how many days were data available? b) What does “specific locomotion behaviors” mean? c) Eating times 1 and 2 have the same descriptions. d) What are “fear bites”? e) Eating chews 1 and 2, check definitions. f) Bolus definition? Isn’t bolus used for feed? g) Chews per minute/per bolus definitions are hard to follow. h) Activity and activity change: how do you define “specific recording periods”?

We corrected - Table 1. During this study, there were registered parameters [9].

Parameter

Description

Registration source

Milk yield (MY)

(kg/session)

Average of milk yield per milking session

DeLaval milking system

Other activity time (OAT)

(min/h)

The time that cows spend engaged in activities other than feeding, rumination, or specific locomotion behaviors

RWS

Rumination time (RT)

(min/h)

Duration of ruminating chews, with up to 5-second chew breaks

RWS

Eating time 1 (ET1) (min/h)

Feeding time with head position down within chosen summary interval

RWS

Eating time 2 (ET2)

(min/h)

Feeding time with head position up within chosen interval

RWS

Drinking time (DT)

(min/h)

Drinking duration, including up to 5-second breaks in between drinks

RWS

Other chews (OC)

(n/h)

Total amount of mastication chews and fear bites during eating

RWS

Rumination chews (RC)

(n/h)

Chewing using the mouth during rumination to mechanically break up regurgitated material into smaller pieces

RWS

Eating chews 1 (EC1)

(n/h)

Number of feeding chews with head position down within chosen summary interval

RWS

Eating chews 2 (EC2)

(n/h)

Number of feeding chews with head position up within chosen summary interval

RWS

Drinking gulps (DG)

(n/h)

The cumulative number of swallows during the drinking process

RWS

Bolus (B) (n/h)

Number of regurgitated boli within the chosen summary interval

RWS

Chews per minute (CPM)

(n/min)

Chewing movements occurring during rumination following regurgitation per minute.

RWS

Chews per bolus (CPB)

(n/boli)

Chews made when ruminating in between regurgitating and ingesting one bolus

RWS

Walking time (WT)

(min/h)

The total amount of time spent walking in a specific recording period expressed as minutes

RWS

Up time (UT)

(min/h)

Time spent eating with the head up

RWS

Down time (DT)

(min/h)

Time spent eating with the head lowered

RWS

Activity change (AC) (n/h)

Activity changes count within the summary time frame

RWS

Rev: L170-172: You need to name the software used for milk recording, here.

Auth: We deleted results of milk production, because it was not aim.

Rev: L186: Environmental or cow temperatures?

Auth: We deleted “temperature average” because we did not measure it.

Rev: L194-195: This implies that some variables were not normally distributed; is this true? If yes, how were those variables analyzed? Are SCK, SCA and H really variables? If this is a syntax issue, please be careful!

Auth: We corrected to – “The mean values of the variables SCK, SCA, and H were compared using the Student's t-test”

Rev: L197-198: Besides the statistical controversies, do you have such results available? Overall, one has to assume what the statistical analysis was; this is not a clear description.

Auth: We deleted this information because these parameters were not analyzed. 

RESULTS

Rev: L202: Lactose? 

Auth: We deleted “lactose”

Rev: L207: TABLE 2: The use of superscripts A, B and C is quite unusual, to say the least.

Auth: We are using such superscripts  - A, B,C for descriptive statistics. This is another one of our papers in which we use this - https://www.mdpi.com/2076-2615/13/20/3293#B20-animals-13-03293

Rev: L214-216: Milk yields are obviously per “milking session”. What an unusual way to report milk yields! Still, about how many recordings do we talk about? Please, consider that no lactose data are presented.

Auth: We corrected milk yield to – “Average milk yield per milking session (MY)” And these sentences - “Average milk yield per milking session (MY). Based on our findings, a statistically significant (p<0.001) reduction in MY was observed in cows afflicted with subclinical ketosis (SCK) when compared to healthy cows. Specifically, within the SCK group, MY was found to be 11.78% lower than in the healthy group. The mean MY in the SCK group was 12.57 (±3.57), whereas in the healthy group, it averaged 14.39 (±3.05). In cows with subclinical acidosis (SCA), the MY mean was 14.25 (±4.65)”

Rev: L222-225: “In the Subclinical Acidosis (SCA) group, OAT was 20.6% higher than in the healthy group. The mean OAT in the SCK group was 35.23 minutes (±17.24), while in the healthy group, it averaged 29.21 minutes (±16.27), and in the SCA group, it was 29.17 minutes (±14.26)”. Is 29.17 min 20.6% higher than 29.21 min?

Auth: We deleted this sentence because it was a mistake and statistically unsifnificant. We changed this paragraph to – “Other activity time (OAT). Regarding OAT, our findings indicated a statistically significant increase in OAT among cows with Subclinical Ketosis (SCK), with a p-value less than 0.001. The mean OAT in the SCK group was 35.23 minutes (±17.24), while in the healthy group, it averaged 29.21 minutes (±16.27), and in the SCA group, it was 29.17 minutes (±14.26)”

Rev: L233-235: “The mean RT in the SCK group was 15.16 hours 233 (±0.54), while in the healthy group, it averaged 18.37 hours (±0.47), and in the Subclinical 234 Acidosis (SCA) group, it was 19.40 hours (±0.7)”. Aren’t these rumination times extremely high? Can this be correct? If you add RT, OAT, ET1 and ET2, it is more than 24 hours. Per day, I assume.

Auth: We corrected whole results section and we clarified units of parameters.

Rev: L239-335: Without proper definitions for most of them, it is impossible to review them. What is an “activity unit”?

Auth: We corrected  - Table 1. During this study, there were registered parameters [9] and added following definitions-

Eating time 1 (ET1) (min/h)

Feeding time with head position down within chosen summary interval

Eating time 2 (ET2)

(min/h)

Feeding time with head position up within chosen interval

Eating chews 1 (EC1)

(n/h)

Number of feeding chews with head position down within chosen summary interval

Eating chews 2 (EC2)

(n/h)

Number of feeding chews with head position up within chosen summary interval

Bolus (B) (n/h)

Number of regurgitated boli within the chosen summary interval

Rev: L336-352: No surprise, really, regarding correlations reported. I need the bolus definition.

Auth: We deleted this information because it does not present any new knowledge. 

DISCUSSION:

Rev: L356-357: Lactose, again.

Auth: We deleted “lactose” and corrected these sentences – “The present study aimed to investigate the effects of subclinical acidosis and ketosis on parameters related to locomotor behavior, rumination, eating. We examined changes in feeding, rumination, and locomotor behavior in cows with subclinical acidosis and subclinical ketosis to test this theory”

Rev: L360-366: These are results; delete.

Auth: Deleted

Rev: L368-370: Any reference for SCK frequently turning clinical?

Auth: We added reference – “Subclinical ketosis frequently goes undetected until it progresses to the clinical stage, despite its widespread occurrence in numerous dairy herds [13]”

Rev: L375-403: All these could be OK, if RT recorded were normal. I would not dare to consider a cow ruminating for 15 hours per day as a sick one, even if others were ruminating for 18 hours. For how long were cows found to be ruminating in other studies and with other equipment? What about lactating cow time budgets?

Auth: We corrected units of all measured parameters, including RT -  (min/h).

Rev: L404-437: This is a text quite difficult to follow. It must be structured correctly.  The sentence in L405-406 makes no sense. SCA and SCK are all mixed up. Without proper definitions on hand, this is an impossible task. Again, what is the “unit” reported? What other researchers report on these issues? How do your results compare to those of others.

L438-451: A long text, just to support findings reported in this and in the previous study. Avoid repeating results. This is the place to report and discuss what others found (not in L484). Shorten and restructure this section.

L451-469: Most of this is “introduction” stuff. Delete or remove. Text in L457-458 does relate to comment above. Early detection of diseases seems to be a recurring issue for the authors.  They provide no such findings themselves, though. What is your point here? Early detection is feasible through behavioral changes? What is your contribution? Do you report changes before any other method of diagnosis?

L470-483: This is an attempt to explain reduced activity. Fair, but with lot of speculation present. Useful only if using literature correctly.

L484-498: Blend this part of the manuscript with the discussion text of corresponding findings.

Auth: We corrected the whole discussion section and restructured it. Also, we added new subsections:

4.1. Alterations in rumination, eating behavior in dairy cows affected by subclinical ketosis and subclinical acidosis. 

4.2. Alterations in drinking behavior in dairy cows affected by subclinical ketosis. 

4.3. Alterations in locomotion behavior in dairy cows affected by subclinical ketosis and subclinical acidosis

Rev: L499-514: Nothing new, nothing useful. Delete or replace with more relevant discussion. 

Auth: We deleted this information from results and discussion sections.

CONCLUSIONS

Rev: L518-521: “In conclusion, our findings strongly support the notion that SCK exerts a significant influence on various aspects of dairy cow health and productivity, including milk production, eating behavior, rumination, body condition, and locomotion”. Did you really find a link between SCK and dairy cow health? What you describe is behavioral changes, not health issues.

Auth: We corrected conclusion section following – “In this study, we analyzed alterations in rumination, drinking eating and locomotion behavior in dairy cows affected by SCK and SCA. In the case of SCK, we found a reduction in RT (duration of ruminating chews with up to 5-second chew breaks) of 17.5%, eating time 1 (feeding time with head position down within the chosen summary interval) of 16.5%, and eating time  2 (feeding time with head position up within chosen interval) of 28.6, in other chews (total amount of mastication chews and fear bites during eating) of 13.8%, in rumination chews (chewing using the mouth during rumination to mechanically break up regurgitated material into smaller pieces) of 16.2%, in eating chews 1 (number of feeding chews with head position down within the chosen summary interval), 19.6%; and in eating chews 2 (number of feeding chews with head position up within the chosen summary interval), 38.1%; in chews per minute (chewing movements occurring during rumination following regurgitation per minute), 16.7%; and in bolus (number of regurgitated boli within the chosen summary interval), 17.0%. Regarding locomotion behavior, we found a reduction of 27.4% in walking time (the total amount of time spent walking in a specific recording period expressed as minutes), 13.78% in activity change (activity changes count within the summary time frame), and an increase of 19.8% in other activity time (the time that cows spend engaged in activities other than feeding, rumination, or specific locomotion behaviors).

During SCA, we found reductions in eating time 2 of 36.8%, eating chews 2 of 38%, and waking time of 19.1%.

Our findings suggest that, by using innovative technologies such as RWS, we can see that SCK and SCA had a significant influence on various aspects of dairy cow behavioral changes, including eating, rumination, and locomotion behavior. These data can be used for creating algorithms based on registered eating, rumination, and locomotion behavior parameters”

Rev: L529-531: “Furthermore, this study underscores the necessity for enhanced strategies aimed at preventing and mitigating subclinical metabolic disorders within the dairy farming industry”.  Just because behavior changes?

Auth: We deleted this sentence.

Round 2

Reviewer 1 Report

Comments and Suggestions for Authors

The manuscript is clearer and much improved. However, you have still reiterated to one decimal place the results in the discussion section. 

Reviewer 2 Report

Comments and Suggestions for Authors

The manuscript has undergone many meaningful revisions and requires formatting changes according to the requirements of the magazine